# Applying ChatGPT to tackle the side effects of personal learning environments from learner and learning perspective: An interview of experts in higher education

XiaoShu Xu[1]*, XiBing Wang[2,3], YunFeng Zhang[4], Rong Zheng[3,5]*

**1** School of Foreign Studies, Wenzhou University, Wenzhou, Zhejiang, China, **2** School of Information Engineering, Yunnan Vocational College of Mechanical and Electrical Technology, Kunming, Yunnan, China, **3** Centre for Portuguese Studies, Macao Polytechnic University, Sé Freguesias, Macao, China, **4** School of Education, Baoshan University, Baoshan, Yunnan, China, **5** Graduate School, Stamford International University, Suan Luang, Bangkok, Thailand

* lisaxu@wzu.edu.cn (XSX); 2203080007@students.stamford.edu (RZ)

## Abstract

This paper investigates the capacity of ChatGPT, an advanced language model created by OpenAI, to mitigate the side effects encountered by learners in Personal Learning Environments (PLEs) within higher education. A series of semi-structured interviews were conducted with six professors and three Information and Communication Technology (ICT) experts. Employing thematic analysis, the interview data were assessed, revealing that the side effects stemming from the learner and learning perspectives could be primarily categorized into cognitive, non-cognitive, and metacognitive challenges. The findings of the thematic analysis indicate that, from a cognitive standpoint, ChatGPT can generate relevant and trustworthy information, furnish personalized learning resources, and facilitate interdisciplinary learning to fully actualize learners' potential. Moreover, ChatGPT can aid learners in cultivating non-cognitive skills, including motivation, perseverance, self-regulation, and self-efficacy, as well as metacognitive abilities such as self-determination, self-efficacy, and self-regulation, by providing tailored feedback, fostering creativity, and stimulating critical thinking activities. This study offers valuable insights for integrating artificial intelligence technologies to unleash the full potential of PLEs in higher education.

**Data Availability Statement:** The dataset is available at the following DOI: 10.6084/m9.figshare.22303441.

## 1. Introduction

Personal Learning Environments (PLEs) is an educational approach that enables students to utilize social media to enhance self-regulated learning in both formal and informal pedagogical settings [1, 2]. The purpose of PLEs is to equip students with the ability to organize the resources, social interactions, and information produced in the teaching-learning process [3, 4]. Research has found that there is a correlation between the use of PLEs and improved student autonomy and decision-making capacity, which can reduce reservations in performing

**Funding:** The author(s) received funding from the National Social Science Foundation of China for Education General Program (BGA210054) for this work. The First Author Xiaoshu Xu received this funding.

**Competing interests:** The authors have declared that no competing interests exist.

tasks [5]. Additionally, self-efficacy, a malleable element [6], can be enhanced over time through the use of virtual or traditional tools and strategies integrated into PLEs.

However, the development and implementation of PLEs can result in unintended negative consequences, referred to as side effects. These side effects can have an impact on both the learners and the learning process, leading to reduced learning outcomes or hindering the achievement of educational goals. Recent literature has identified various forms of side effects in education, including distractions, information overload, and reduced interaction between learners and teachers caused by the use of technology in the classroom [7, 8]. Similarly, challenges related to privacy, data security, and learner autonomy can arise from the implementation of PLEs [9, 10]. Pedagogical approaches can also create side effects, such as anxiety, stress, and academic disengagement caused by the use of high-stakes testing or competitive grading systems[11, 12].

To mitigate the potential side effects of PLEs, we aim to explore how ChatGPT, a language model that provides personalized responses to learners, can address the potential side effects of PLEs from the learner and learning perspective. We plan to conduct interviews with experts in higher education to discuss the potential side effects of PLEs, the benefits of using ChatGPT, and how it can enhance the learning experience. By doing so, we hope to provide insights into how AI technologies such as ChatGPT can be leveraged to improve the effectiveness and efficiency of PLEs while minimizing potential side effects.

## 1.1 Concept and development of Personal Learning Environments (PLEs)

Personal Learning Environments (PLEs) provide learners with the ability to create and share their own content, work collaboratively with others, and access resources from anywhere and at any time through incorporating different digital tools and resources, such as social media, blogs, wikis, and other web-based applications. PLEs offer a flexible and dynamic learning experience that can be tailored to suit the needs and interests of individual learners, resulting in increased engagement and improved learning outcomes. According to Castañeda's [13] analysis of the PLE literature published during the period 2010–2020, two critical areas of research have emerged: pedagogical practices and self-regulated learning. The notion of PLEs has been linked with the creation of innovative learning models that challenge traditional educational structures and prioritize the learner, granting greater autonomy and placing them at the center of the learning process. In this regard, teachers are viewed as facilitators and guides. Meanwhile, research on self-regulated learning suggests that PLEs is primarily examined from a pedagogical perspective, albeit with subtle variations. Moreover, digital tools, and Web 2.0 services are considered as instruments for learners to manage their learning and enhance cognitive skills associated with self-regulated learning.

## 1.2 Side effects of PLE in higher education from a learner and learning perspective

Studying the side effects of education is crucial for providing learners with an optimal educational experience. Metcalf [14] research suggests that understanding the potential negative consequences of educational interventions can help educators and policymakers make informed decisions about the design and implementation of educational programs. In terms of the side effects of PLEs, one of the primary challenges identified is the integration of PLEs with formal learning management systems (LMS). This challenge arises due to the differences in structure and functionality between PLEs and LMSs, which may lead to issues with interoperability and data exchange [1]. Kop and Hill [15] identified the gap between individuals with access to digital technologies and those without, as well as the quality and reliability of the

information available within PLEs. Fiedler and Väljataga [16] highlighted the challenge as the lack of a clear and consistent definition of PLEs. Blaschke [17] discussed the challenge of incorporating social media into existing formal educational systems. Wilson et al. [18] addressed the issue of control and ownership of learning within PLEs. While PLEs aim to empower learners by giving them control over their learning experience, this may conflict with traditional educational systems, where control is often held by institutions and educators.

From the learner and learning perspective, this research reviewed the side effects and challenges of PLEs, and summarized them into the following three categories: cognitive skills, non-cognitive skills, and meta-cognitive skills. Regarding cognitive skills, learners require sufficient digital literacy to use PLEs effectively. For instance, Lee and Meina [19] and Attwell [20] suggest that keeping up with new technologies and developing necessary digital literacy skills can be difficult for learners due to the fast pace of technological change. Furthermore, the abundance of information available online can be overwhelming, making it challenging for learners to distinguish credible sources and effectively utilize digital tools [1, 15].

In terms of non-cognitive skills, it includes motivation, perseverance, self-regulation, and self-efficacy. According to Kim [21], learners' motivation and emotions are critical factors for success in PLEs. However, effectively measuring and utilizing non-cognitive skills in PLEs remains a challenge. Routledge, Humphries and Kosse [22, 23] notes that non-cognitive skills are difficult to measure and develop, which poses a challenge to their effective utilization in PLEs.

As for meta-cognitive skills, it includes self-determination, self-efficacy, and self-regulation, which involves setting personal learning goals, selecting and organizing resources, and evaluating their learning outcomes. Learners who lack these skills may face difficulties in managing their PLEs effectively [16]. Rodman [24] suggest that PLEs require learners to be proactive in their learning, taking ownership of their learning outcomes. Developing learners' self-efficacy is another critical challenge in PLEs. Jeske et al. [25] argue that learners must feel confident in their abilities to succeed in a personalized learning environment. Lastly, learner self-regulation and management of Web 2.0 tools pose challenges in the development and implementation of PLEs. Lim and Newby [26] highlight the importance of learners' self-regulation in the effective use of Web 2.0 tools, emphasizing the need for learners to manage their online learning activities effectively.

To succeed in PLEs, learners need to have a range of cognitive, non-cognitive, and meta-cognitive skills. While digital literacy, motivation, self-regulation, and self-efficacy are important for success, developing and utilizing these skills effectively can be challenging for both learners and educators.

## 2. Methods

The study received approval from the Wenzhou University Review Board, bearing the approval number IORG/IRB #202303. For data collection, online semi-structured interviews were carried out during 20–27 February, 2023, utilizing the Tencent Conference platform. The interviews were conducted in English and lasted between 25 to 30 minutes. Following their participation, each interviewee was sent a letter of appreciation acknowledging their valuable contribution. Throughout the research, the team was deeply conscious of ethical standards and was committed to upholding them. Before proceeding with interviews, written informed consent was secured from every participant via email, reflecting our commitment to respecting their choices. Strict confidentiality was maintained regarding the participants' information, ensuring their anonymity and privacy. Their data was handled with utmost discretion, and their identities were never mentioned in any resulting publications or reports.

The interview questions were designed to explore the potential of ChatGPT in addressing the side effects of PLEs from the perspectives of the learner and learning. The interviews were audio-recorded and transcribed for further analysis. The open-ended nature of the questions allowed participants to provide detailed responses based on their expertise and experience. After the interview, all the interviewees would receive a thank-you letter for their participation and contribution to the study. Our adherence to these ethical standards underscored our commitment to conducting the study with respect and integrity.

Nonetheless, it is plausible to anticipate that the modest sample size might influence the comprehensive comprehension of the research inquiry. Simultaneously, relying solely on expert interview methodology could circumscribe the scope and richness of the information amassed. In subsequent investigations, scholars might contemplate employing supplementary data collection techniques, such as questionnaires or observations, to garner a more holistic understanding of learners' experiences with PLEs and ChatGPT.

## 2.1 Participants

The experts were chosen using a purposive sampling method, where individuals with relevant expertise and experience were identified and invited to participate in the study. Selection criteria were pre-defined to ensure that only participants with the necessary knowledge and experience were included. The selection criteria for the ICT expert interviewees and higher education professor interviewees are as follow (Table 1):

For this study, a total of nine experts were selected, comprising of three ICT experts and six higher education professors. The ICT experts had a considerable amount of experience in platform development, with 10–20 years of experience, aged between 40–50, and two of them held Ph.Ds. and one held MA in relevant fields. They were from various higher education institutions in different parts of China, with two being male and one female. On the other hand, the six higher education professors had a clear understanding of PLEs, with 5–20 years of teaching experience, and were between the ages of 35–55. Four of them held PhD degrees and two had master degrees. Four of them were females. Two of them were lecturers, another two were associate professors, and the rest two were professors. All of them had experience in teaching using Learning Management Systems (LMS), and three of them had developed PLE platforms that had been operational for 5–8 years (see Table 2).

## 2.2 Research instrument

The research instrument used for this study was an interview using ten open-ended interview questions (see S1 Appendix). The questions aimed to gather data on the experts' understanding of PLE and ChatGPT, their predictions on the application of ChatGPT to PLE, their

**Table 1. Selection criteria for the ICT expert interviewees and higher education professor interviewees.**

| ICT expert interviewees | Education professor interviewees |
|---|---|
| Have at least five years of platform development experience in higher education | With at least five years of higher education teaching experience |
| Have experience in developing and implementing online learning platforms, especially personalized learning platforms. | Have experience in teaching using Learning Management Systems (LMS), Blackboard or some other similar online platforms |
| Have expertise in developing ICT tools that can be applied to address the challenges that learners face in PLEs. | Have a clear understanding of PLEs |
| Have master or Ph.D. degree | Have Master or Ph.D. degrees |
| Have both male and female experts | Have both male and female professors |

**Table 2. Personal information of interviewees.**

| No. | Gender | Teaching experience | Age | Degree | position |
|---|---|---|---|---|---|
| Expert interviewee 1 | Male | 13 | 43 | Ph.D | ICT expert |
| Expert interviewee 2 | Male | 20 | 50 | MA | ICT expert |
| Expert interviewee 3 | Female | 10 | 40 | Ph.D | ICT expert |
| Professor interviewee 1 | Female | 5 | 35 | MA | Lecturer |
| Professor interviewee 2 | Male | 20 | 55 | Ph.D | Professor |
| Professor interviewee 3 | Female | 11 | 40 | MA | Associate Professor |
| Professor interviewee 4 | Male | 6 | 37 | MA | Lecturer |
| Professor interviewee 5 | Female | 19 | 53 | Ph.D | Professor |
| Professor interviewee 6 | Female | 17 | 48 | Ph.D | Associate Professor |

experience with challenges in implementing PLEs in higher education, and their suggestions on using ChatGPT to address the side effects of PLE from learner and learning perspectives.

The interview questions were divided into three categories. The first set (Questions 1–3) aimed to understand the experts' definitions of PLE, their experience with PLE platforms, and their familiarity with ChatGPT. The second set (Questions 4–5) focused on the potential benefits and challenges of incorporating ChatGPT in PLEs, as well as the effectiveness of ChatGPT in addressing the side effects of PLEs. The third set (Questions 6–7) explored the experts' experience with the challenges of PLE implementation in higher education, including the challenges encountered by all stakeholders. Finally, Questions 8–10 were open-ended, asking the experts to provide suggestions on how to use ChatGPT to develop learners' cognitive, non-cognitive, and metacognitive skills and support them in PLEs. The interview questions were reviewed by two experts in the field of Personal Learning Environments to assess their relevance, clarity, simplicity, and comprehensibility.

## 3. Results and data analysis

Thematic analysis is a flexible and systematic approach that can be used for a variety of research questions and data types, including interview transcripts. The method provides a rigorous and comprehensive understanding of the data, allowing researchers to identify important patterns and themes that may not be immediately apparent [27]. In order to analyze the data gathered from the interviews, two researchers were enlisted to participate in the coding process, adhering to the methodology put forth by Elo and Kyngäs [28]. The initial step involved the first author reviewing all the textual content and generating a comprehensive list of preliminary codes. These codes were subsequently organized into various themes. The first and second authors then convened to discuss and refine the codes and themes, resulting in some themes being reshaped. The two researchers conducted their own separate thematic analysis, referencing the established list of codes and themes. Once all the codes and themes were identified by both researchers, an extensive comparison and discussion ensued. Definitions were assigned to all codes, while some themes were reclassified. The researchers then conducted a second round of coding based on the revised themes. Following subsequent deliberation, the two researchers were able to identify over 90% of similar themes, codes, and references (refer to Table 3).

### 3.1 The first set of interview questions

The responses from the first set of questions (Questions 1–3) provide valuable insights into the experts' understanding of Personal Learning Environments (PLEs) and ChatGPT. All three

**Table 3. Themes and sub-themes extracted from the opinions of the interviewees.**

| Themes | Subthemes |
|---|---|
| PLEs & ChatGPT understanding | take charge of their learning<br>personalize learning experience<br>increase engagement and motivation<br>potential-realization<br>conversational AI |
| PLE experience | significant experience<br>positive experience<br>critical challenges<br>PLE design<br>conflicts with formal education |
| ChatGPT familiarity | Familiar<br>Some experience<br>Benefits to apply ChatGPT in PLE<br>personalized and timely feedback and support<br>ChatGPT design<br>Challenges to apply ChatGPT |
| ChatGPT incorporate PLEs | system design<br>interface design<br>effectively integrated<br>formative assessment<br>loss of personal interaction and engagement<br>complex and time-consuming<br>significant resources and technical expertise |
| Benefits of ChatGPT incorporate PLEs | increased personalization<br>more flexibility and better quality of learning material<br>user-friendly interface<br>improved learning engagement<br>collaborative learning<br>better outcome<br>increased learning motivation<br>increased efficiency of teaching and learning |
| One's experience of PLEs implementation challenges | resistance to change<br>integration issues<br>effectiveness of PLEs<br>accessibility<br>ICT literacy<br>the lack of training<br>ethical issues |
| Challenges of PLEs implementation involve all stakeholders | design PLEs platform for diverse needs<br>provides personalized support and feedback<br>accessible and inclusive<br>provide adequate training and support<br>system integration<br>security and maintenance of PLEs<br>adjustment<br>shifting educational philology |
| ChatGPT addresses these challenges | provide real-time support and guidance<br>credible sources of information<br>provide personalized feedback, recommendations, sources<br>adapt to individual needs and abilities<br>create formative assessments<br>time management and organization<br>maintain learner motivation and engagement<br>help develop self-regulated learning |

(*Continued*)

**Table 3.** (Continued)

| Themes | Subthemes |
|---|---|
| ChatGPT develops non-cognitive skills | timely and targeted feedback<br>help to stay motivated and engaged<br>develop self-regulation skills<br>foster perseverance and self-efficacy<br>overcome challenges<br>develop problem-solving skills |
| ChatGPT developing metacognitive skills | help managing Web 2.0 tools<br>encourage learners to take an active role<br>prompt to reflect on learning<br>a better understanding of strengths and weaknesses<br>develop self-awareness<br>giving learners more control |

experts agreed that PLEs were digital environments that empowered learners to take charge of their learning and customize their learning experiences according to their preferences and needs. Similarly, the experts concurred that ChatGPT was a conversational AI platform that could be seamlessly integrated into PLEs to offer learners personalized and timely feedback and support. For instance, expert interviewee 2 said:

> *"When you bring Chat into PLEs, it can interact with learners, answer their questions, and guide them through their learning. Like having a digital tutor 24/7".*

All three experts were familiar with ChatGPT technology and have used it in the development of PLEs. They agreed that ChatGPT could be an effective way to provide learners with personalized and timely feedback and support to enhance their Self-directed Learning and digital literacy skills. For example, in answering question 2, expert interviewee 3 mentioned that:

> *"I've seen how PLEs can help students take control of their learning, find resources and digital tools that fit their needs, and engage more deeply with the material. It's been very rewarding to see students become more active and self-directed in their learning."*

Additionally, they emphasized the importance of designing ChatGPT to be engaging and user-friendly to maximize its effectiveness.

Regarding the six professors who were interviewed, five of them described PLEs as digital environments that enable learners to personalize their learning experiences and access a broad range of digital resources and tools. This definition aligns with the prevailing understanding of PLEs within the field of education. For instance, in answering question 2, professor interviewee 3 pointed out that:

> *"They have the freedom to choose the digital resources and tools that best suit their learning styles and preferences. This customization empowers them to take ownership of their education and fosters a sense of responsibility for their learning outcomes."*

The interviewees also recognized the potential of PLEs to increase learners' engagement and motivation, which were crucial factors in the learning process. Two professors highlighted the possibility of potential realization education with the development of PLEs.

In terms of their familiarity with ChatGPT, four professor interviewees reported having plenty of experience in using ChatGPT and the rest two reported had tried it in academic

research. All experts recognized the potential benefits of incorporating ChatGPT in PLEs. They highlighted its ability to provide learners with personalized and timely feedback, support, and guidance.

However, two experts also raised concerns about the challenges of ensuring the accuracy and appropriateness of the feedback provided by ChatGPT. Three of them emphasized the importance of designing conversational AI to be engaging and user-friendly to maximize its effectiveness. For instance, expert interviewee 1 said:

*"Making it engaging and user-friendly can truly amplify its effectiveness. You want to make sure that learners find it easy to interact with and feel comfortable asking questions. That's when the real learning happens."*

### 3.2 The second set of questions

**3.2.1 Question 4: Do you think there is a possibility to incorporate ChatGPT technology into PLEs?.**   Three ICT experts explained the way to incorporate ChatGPT into PLEs from two perspectives, including system design and interface standpoint. From the system design perspective, expert interviewee 2 said:

*"To effectively integrate ChatGPT technology into PLEs, a rigorous system design approach is necessary to ensure that performance and user experience are not compromised. This involves conducting a thorough analysis and evaluation of both the PLEs and ChatGPT technology to identify any potential compatibility issues that may arise."*

From the interface standpoint, expert interviewee 1 said:

*"In order to smoothly incorporate ChatGPT technology, it's essential to create a user-friendly interface design. One approach is to develop a dedicated ChatGPT plugin that can be integrated with the PLE interface. . . . . . . The plugin provides a seamless and easily accessible interface for learners to engage with ChatGPT."*

Expert interviewee 3 added the answer from users' perspective:

*"In order to ensure that the feedback and support provided are relevant and meaningful, the interface design must take into account the users' learning objectives and the context of the learning activity. It can maximize the effectiveness of the feedback and support provided, and enhance the overall learning experience."*

The six professors who were interviewed had different opinions regarding the use of ChatGPT in PLEs. Half of them believed that this technology could be effectively integrated into PLEs, providing personalized feedback and support for learners and creating formative assessments (e.g., quizzes or short writing tasks), which would enhance the quality of learning and promote a more efficient and effective learning environment.

However, the other three expressed concerns about the potential loss of personal interaction and engagement between students and instructors, which were crucial for effective learning. They also highlighted that the integration process could be complex and time-consuming, requiring significant resources and technical expertise. Additionally, there were concerns about the quality and accuracy of the feedback provided by ChatGPT, which could negatively impact learning outcomes. The response emphasized the importance of carefully evaluating the benefits and drawbacks of using ChatGPT in PLEs before deciding to integrate this technology.

**3.2.2 Question 5: In your opinion, what are the benefits of incorporating ChatGPT technology into PLEs?.**    Three ICT experts indicated the benefits from three perspectives, including increased personalization, more flexibility and better quality of learning material, and user-friendly interface. For instance, expert interviewee 1 said:

*"An important benefit of ChatGPT is its ability to provide customized support and feedback to learners, and it can improve the learning process to be more effective and efficient. Integrating ChatGPT into PLEs can give learners timely feedback and fully explore conversational AI features. Learners can improve their weak areas and be more engaged in the interactive learning platform. They will finally be more successful."*

Expert interviewee 2 said:

*"Apply ChatGPT to PLEs can increase flexibility in accessing learning materials, and allow learners to learn at their own pace. This flexibility facilitates learners to be more engaged in learning, and it will increase student retention and success."*

Expert interviewee 3 said:

*"The creation of a specialized ChatGPT plugin can guarantee a smooth incorporation with the PLEs interface. This can offer learners a user-friendly and straightforward platform, and they will be more engaged with the technology."*

The six professor interviewees expressed their expectations and concerns on this question. Most of them were positive about the benefits, such as efficient collaborative learning, improved learning engagement, better outcome, increased learning motivation, and increased efficiency of teaching and learning. For instance, professor interviewee 2 said:

*"ChatGPT can significantly enhance student engagement. It offers them immediate personalized feedback and support, and assists them to find out their strengths and weaknesses, and, in turn, improve their learning achievements."*

Professor interviewee 5 said:

*"The integration of ChatGPT technology into PLEs can enhance learning motivation. It can facilitate collaborative learning by supporting group discussions, brainstorming sessions, and project work. It can act as a mediator, offering suggestions, guiding discussions, and helping learners work together effectively."*

Professor interviewee 6 said:

*"ChatGPT technology can lead to increased learning efficiency. It can decrease the workload for instructors. As a result, instructors can devote more time to personalized teaching methods. This can improve the overall quality of education."*

## 3.3 The third set of questions

**3.3.1 Question 6: Have you experienced any challenges when implementing PLEs in higher education? If so, can you describe them?.**    The three ICT experts mentioned several

challenges that need to be addressed when implementing PLEs in higher education. Firstly, they acknowledged the potential resistance to change from faculty and students accustomed to traditional classroom settings. Secondly, integration issues with existing university systems could arise, requiring careful planning and coordination. Thirdly, ensuring that PLEs are effective in promoting student learning and engagement is crucial. The experts emphasized the importance of designing a platform that can meet diverse student needs and provides personalized support. Additionally, there may be challenges in making the platform accessible to all students, including those with disabilities or limited technology access. Overall, the experts highlighted the need for careful planning and stakeholder engagement to ensure successful implementation of PLEs in higher education.

The six professor interviewees indicated the challenges from three perspectives, including ICT literacy, technology accessibility, the lack of training, and ethical issues. For instance, professor interviewee 3 said:

*"One of the primary challenges to successful integration is the varying levels of ICT skills among students and faculty. This can negatively impact the platform's efficacy. So, it is critical to offer training and support to all users. We have to ensure that all of them are comfortable and when using the platform."*

Professor interviewee 2 further pointed out that:

*"some students may lack consistent access to the necessary technology, including computers and reliable internet access. This digital divide can make it difficult for these students to fully participate in a PLE."*

Professor interviewee 4 said:

*"Lack of resources or training for faculty members will hinder the quality of the learning experience for students. It is important to offer appropriate support and training to ensure the successful implementation and utilization of PLEs."*

Professor interviewee 1 added:

*"It's also important to continuously update the training as the technology evolves and as we gain more insights into effective teaching and learning strategies in digital environments. However, developing and providing this training requires a significant investment of time and resources, which can be a challenge in itself."*

**3.3.2 Question 7: What do you think are the challenges that all stakeholders, including students, instructors, and administrators, face when implementing PLEs in higher education?.** All three ICT experts and the six professor interviewees expressed similar opinions regarding the challenges of implementing PLEs in higher education. They identified the challenge of designing a platform that catered to the diverse needs of students and provided personalized support and feedback as the most significant challenge. They emphasized the importance of careful planning and design to ensure that the platform was accessible and inclusive for all students, including those with disabilities or limited technology access. Additionally, they highlighted the importance of providing adequate training and support for instructors to effectively incorporate PLEs into their teaching practice, which may require additional resources and professional development.

The ICT expert interviewees pointed out that administrators may face challenges with integrating the PLE with existing university systems and technologies, which can cause technical difficulties and limit access to certain resources. They may also face challenges with securing funding and resources to support the implementation and ongoing maintenance of the PLE. For instance, expert interviewee 2 pointed out:

*"One of the challenges that administrators may encounter is the funding and resource allocation. The implementation of PLEs require significant investments in technology and infrastructure, which may be costly. And, it also requires changes in policies and procedures to support PLEs."*

The professor interviewees highlighted that students may face challenges with adjusting to a new learning environment and may require additional support and guidance to effectively use the PLE. They may also face challenges with accessibility and may require accommodations for disabilities or limited access to technology.

Besides, professor interviewees 3 and 5 highlighted the challenges of shifting educational philology among teachers and learners, from score-oriented to ability building. For instance, interview 3 said:

*"In order to successfully integrate PLEs into the classroom, teachers may need to adjust their teaching roles from being traditional lecturers to facilitators of learning. This shift may require additional training and support to ensure that teachers feel confident and comfortable in their new role."*

Interview 5 said:

*"In PLEs, students may encounter difficulties adjusting to the new learning approach, particularly if they are accustomed to conventional classroom settings. They may require additional assistance to cultivate the necessary abilities to excel in such an environment, including time management, self-motivation, and self-directed learning."*

In terms of administrators, the professor interviewees mainly pointed out the following three challenges, including: infrastructure and technical support, that is administrators need to ensure that the institution has the necessary technical infrastructure to support the implementation of PLEs. This includes reliable internet access and user-friendly platforms; second, professor interviewee 4 and 5 mentioned the challenge of resource allocation, which means when implementing PLEs, administrators must strategically allocate resources to guarantee the effective integration of PLEs in higher education; third, professor interviewee 3 and 6 highlighted the data privacy and security issue. professor interviewee 3 said:

*"Protecting student data and ensuring privacy and security are crucial considerations when implementing PLEs. Administrators must address potential risks and ensure compliance with data protection regulations."*

### 3.4 The fourth set of question

**3.4.1 Question 8: How can ChatGPT be used to address the challenges learners face in keeping up with technological advancements and effectively using digital tools to find credible sources of information, particularly in relation to learners' digital literacy**

**skills?.** Two ICT experts explained that ChatGPT could be used to address learners' digital literacy challenges by providing real-time support and guidance as they navigate digital tools and search for credible sources of information. The conversational AI capabilities of ChatGPT allow it to understand learners' queries and provide personalized feedback, recommendations, and resources based on their individual needs and learning preferences. Expert interviewee 3 pointed out that:

*"ChatGPT can suggest various online courses, tutorials, and other resources to help learners enhance their digital literacy skills and keep pace with the latest technological advancements."*

Expert interviewee 2 said:

*"The integration of ChatGPT technology in PLEs enables students to receive timely assistance, it also direct them in using digital tools to find trustworthy sources of information. I believe it can boost their digital literacy abilities and improve their overall learning journey."*

In terms of the professor interviewees, four of them agreed that ChatGPT could be a valuable tool in addressing the challenges learners face in keeping up with technological advancements and effectively using digital tools to find credible sources of information. By leveraging ChatGPT's conversational AI capabilities, students can receive real-time feedback and support to help them navigate the complexities of digital literacy. For example, professor interviewee 1 said:

*"So, with ChatGPT, students can get help in figuring out what sources of information are legit, learn how to check if stuff on the internet is trustworthy, and generally get better at using technology."*

The other two professor interviewees pointed out that ChatGPT can be integrated into PLEs to provide a personalized learning experience that adapts to the individual needs and abilities of each student, helping to promote student engagement and success in the digital age. For instance, professor interviewees 6 said:

*"With ChatGPT, it's like having your own personal research assistant! It can analyze your search terms and suggest relevant sources of information, as well as help you figure out if they're trustworthy or not. Pretty cool, right?"*

Professor interviewee 3 added that ChatGPT can help develop learners' critical thinking skills:

*"ChatGPT can engage students in discussions about evaluating online sources for credibility. By asking critical questions and guiding learners through the process of source evaluation, it promotes critical thinking skill."*

Finally, professor interviewee 4 raised the issue of multimodal learning. He said:

*"ChatGPT can integrate multimedia content, such as videos and images, to enhance learners' understanding of digital tools and information sourcing processes."*

### 3.4.2 Question 9: How can ChatGPT be used to develop non-cognitive skills such as motivation, perseverance, self-regulation, and self-efficacy in learners using PLEs?

Expert interviewees 1 and 3 explained that ChatGPT can provide learners with timely and targeted feedback on their progress, helping them to stay motivated and engaged with their learning. Additionally, ChatGPT can help learners to develop self-regulation skills by providing them with tools and resources to help them manage their time, track their progress, and stay on task. For istance, expert interviewee 2 said:

> "We could totally hook up PLE with AI and machine learning algorithms. This would let the system analyze how each student learns and behaves, and pinpoint where they might struggle with motivation, self-regulation, or feeling confident in themselves." *And he further explained:*

> *"So, basically, the system could look at how each student learns and behaves, and then give them specific help and advice to improve in areas where they might be struggling."*

Three of the professor interviewees stated that ChatGPT technology can be beneficial in fostering non-cognitive skills like motivation, self-regulation, perseverance, and self-efficacy in learners who use PLEs. They explained that the personalized feedback and support provided by ChatGPT could assist students in setting and achieving goals, reflecting on their learning progress, and developing self-efficacy. Furthermore, the real-time assistance and guidance offered by ChatGPT could help students overcome challenges and develop problem-solving skills and perseverance. ChatGPT could also promote self-regulation by empowering students to take ownership of their learning and make informed decisions about their educational journey.

Professor interviewee 2 pointed out the possibility to encourage peer-to-peer interaction through ChatGPT:

> *"When students team up with their classmates using ChatGPT, they can acquire critical social abilities, such as effective communication, collaborative work, and empathy. By creating connections with others in their learning circle, students can also foster a sense of belonging and eagerness to take part more actively in their education."*

Professor interviewee 1 suggested incorporating ChatGPT as a tool for self-reflection and goal-setting:

> *"If students use ChatGPT often, they can keep an eye on how they're doing and see if they're getting closer to their goals. This can help them become more aware of their strengths and weaknesses, and learn how to take charge of their own learning journey."*

### 3.4.3 Question 10: How can ChatGPT be used to develop metacognitive skills such as self-determination, self-efficacy, and self-regulation in learners using PLEs, particularly in terms of their management of Web 2.0 tools and taking an active role in their learning outcomes?

For this question, all the three ICT experts suggested that ChatGPT could be a valuable tool in helping learners develop metacognitive skills by providing personalized feedback and support that helps them take an active role in their learning. For instance, expert interviewee 2 said:

*"One way to do this is by incorporating ChatGPT into PLEs as a virtual learning assistant that guides learners through the process of managing Web 2.0 tools. It can help them develop the skills they need to become effective digital learners."*

Expert interviewee 3 said:

*"ChatGPT can be programmed to provide feedback and guidance on the use of different digital tools and resources. It can help learners to develop the skills they need to use these tools effectively. It can also be used to encourage them to take an active role in their learning, for example, by setting goals, monitoring their progress, and providing feedback on their performance."*

Besides, expert interviewee 1 pointed out that ChatGPT could be programmed to prompt learners to reflect on their learning experiences and evaluate their own progress, helping them to develop a better understanding of their strengths and weaknesses as learners.

Moreover, expert 2 and 3 pointed out that ChatGPT can be programmed to ask students reflective questions about their learning process and progress, such as "What have you learned so far?", "What strategies have you used to complete this task?", or "What can you do differently next time?". By encouraging students to think critically about their learning experiences, ChatGPT can help them develop self-awareness and metacognitive skills.

Finally, as expert interviewee 1 suggested, ChatGPT could be integrated with other digital tools and resources, such as online learning modules, quizzes, and interactive simulations, to provide students with opportunities for self-directed learning and exploration. By giving students more control over their learning experiences, ChatGPT can help them develop self-efficacy and motivation to learn.

As for educator interviewees, two of them suggested that ChatGPT could be used to support the development of metacognitive skills in learners using PLEs by providing a personalized and interactive learning experience. They explained that with ChatGPT, learners could actively engage with the platform and take control of their own learning process. This can be particularly helpful for developing skills related to self-determination, self-efficacy, and self-regulation, as learners can set their own learning goals and monitor their progress toward achieving them. Another two educator interviewees pointed out that ChatGPT could provide learners with guidance and support in using Web 2.0 tools effectively and efficiently. Through its natural language processing capabilities, ChatGPT could help learners identify credible sources of information and evaluate the quality of the content they encounter. This can help learners develop critical thinking skills and become more discerning consumers of digital media.

Educator interviewee 4 said:

*"ChatGPT can help learners stay on track and provide feedback and encouragement throughout the learning process."*

He further explained, *"this can help learners develop a sense of ownership over their learning and build confidence in their ability to achieve their goals."*

Moreover, Educator interviewee 2 and 5 suggested, ChatGPT could facilitate peer-to-peer learning and collaboration, by connecting students with similar interests and goals, and providing opportunities for them to share knowledge and resources. Through fostering a sense of community and social support, ChatGPT can help students develop social and emotional skills, such as empathy, communication, and teamwork.

## 4. Discussion

Based on the analysis of the interview data, learners using PLEs face challenges such as the overwhelming amount of information available online and the need for digital literacy skills to use PLEs effectively. Expert interviewee 2 suggests that ChatGPT can address these challenges by offering learners customized learning materials and instantly presenting relevant and credible information. Simultaneously, research emphasizes the value of proficient information management within PLEs [29]. As expert interviewee 3 notes, ChatGPT can fulfill this demand by supplying appropriate information to its users.

Moreover, prior research underscores the importance of digital literacy skills in online learning environments [30, 31]. This mirrors challenges faced by learners in PLEs, who require substantial digital literacy to utilize the platform optimally. Expert interviewee 3 indicated that ChatGPT can recommend tutorials and additional materials to bolster learners' digital literacy, which resonates with findings from [19, 25, 26], suggesting that ChatGPT's tailored learning resources can enhance digital literacy.

As for non-cognitive skills in applying PLEs, motivation, perseverance, self-regulation, and self-efficacy are proved to be crucial for learners to succeed in online learning [32, 33]. Studies have emphasized the significance of these skills in online learning environment [21, 34, 35]. As the three professor interviewees pointed out that ChatGPT, through personalized feedback and real-time assistance, could enhance learners' non-cognitive skills like motivation and self-efficacy in PLEs, aiding in goal-setting and overcoming challenges. Meanwhile, Nadira et al. [34] suggested that ChatGPT can offer motivational messages and tips to help learners stay motivated and persevere through challenging learning tasks [34].

In terms of metacognitive skills, setting individual learning objectives, resource management, and evaluating outcomes are regarded as crucial for success in PLEs [16]. This mirrors Kim's [36] observation that reflection allows students to be more attuned metacognitively to their cognitive activities, promotes profound comprehension, enables monitoring of acquired knowledge, values the learning journey, and assesses both the learning methodology and students' performance [36]. Besides, Erdogan [37] found that combining cooperative learning with reflective thinking activities can bolster students' critical and metacognitive skills [37]. Similarly, as pointed out by Educator interviewees 2 and 5, ChatGPT, by promoting a community feel via peer-to-peer learning, can aid in developing learners' socio-emotional capabilities such as empathy, clear communication, and collaborative teamwork. Moreover, ChatGPT can guide learners in pinpointing trustworthy information sources and assessing content quality, thus sharpening their critical thinking and making them judicious digital media consumers.

Concerning the roles that teachers or instructors assume in the evolution and application of PLEs in higher education, their primary transition should be from traditional "knowledge deliverers" to "learning facilitators/coaches". They must recognize the paradigm shift towards PLEs in higher education [38] and equip themselves with foundational digital skills, the new educational philosophy [39] (e.g., competence-based learning) and new pedagogical designs, updated Information and Communication Technology (ICT) [40], ensuring they can adeptly support learners in the development of PLEs.

In summary, ChatGPT can be a useful tool to overcome the obstacles encountered by learners in PLEs, and improve their success in digital learning environments. ChatGPT offers personalized resources and feedback, which can aid learners in coping with information overload, enhance their digital literacy, and cultivate critical non-cognitive and metacognitive skills.

## 5. Conclusion and future research

This research reviewed the literature and summarized some of the side effects of PLEs when implemented in the higher education. From the learner and learning perspective, the challenges were categoried into three types, including cognitive, non-cognitive and meta-cognitive. Expert interview was conducted to tackle these challenges. The analysis of expert interviews reveals the potential of ChatGPT in addressing challenges faced by learners using PLEs, particularly in terms of digital literacy and information management. ChatGPT's real-time support and guidance offer personalized feedback, recommendations, and resources tailored to individual needs and preferences. In terms of non-cognitive skill development, ChatGPT can enhance motivation, perseverance, self-regulation, and self-efficacy in PLE users by providing timely feedback, promoting goal-setting, and encouraging self-reflection. Moreover, it fosters peer-to-peer interaction, building critical social skills and a sense of belonging among learners. Regarding metacognitive skills, ChatGPT assists learners in developing self-determination, self-efficacy, and self-regulation by offering personalized feedback, goal monitoring, and guidance in managing Web 2.0 tool. Furthermore, it supports critical thinking, information evaluation, and effective digital media consumption.

In conclusion, ChatGPT holds significant promise in overcoming challenges faced by learners in PLEs, such as information overload, digital literacy, and the development of non-cognitive and metacognitive skills. Its personalized approach and conversational AI capabilities make it a valuable tool for enhancing learner success in digital learning environments.

This study presents certain limitations: initially, the sample comprised only three ICT experts and six professors, possibly not encapsulating the viewpoints of other essential stakeholders in higher education like teachers and students. Additionally, the semi-structured interview format depended on participants' self-disclosures, potentially introducing biases and inaccuracies in their recollections and perceptions.

Regarding forthcoming studies, it's essential to recognize that ChatGPT is an emerging technology with its capabilities continually evolving. As a result, the general understanding of ChatGPT might not be exhaustive. Upcoming research might lean on experimental approaches and surveys to more deeply probe the extent to which ChatGPT can bolster areas like learners' digital literacy, non-cognitive facets, and metacognitive capabilities.

## Supporting information

**S1 Appendix. Interview questions.**
(DOCX)

## Author Contributions

**Conceptualization:** XiaoShu Xu.

**Data curation:** XiaoShu Xu, XiBing Wang.

**Formal analysis:** XiBing Wang.

**Investigation:** YunFeng Zhang, Rong Zheng.

**Methodology:** XiaoShu Xu.

**Project administration:** Rong Zheng.

**Resources:** Rong Zheng.

**Software:** XiBing Wang.

**Supervision:** Rong Zheng.

**Validation:** Rong Zheng.

**Writing – original draft:** XiaoShu Xu.

**Writing – review & editing:** YunFeng Zhang.

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
