## [Decision Letter · Decision Letter 0]

20 Jul 2023

PONE-D-23-08682Applying ChatGPT to tackle the side effects of Personal Learning Environments from learner and learning perspective: an interview of experts in higher educationPLOS ONE

Dear Dr. Zheng,

Thank you for submitting your manuscript to PLOS ONE. After careful consideration, we feel that it has merit but does not fully meet PLOS ONE’s publication criteria as it currently stands. Therefore, we invite you to submit a revised version of the manuscript that addresses the points raised during the review process. Please complete the minor revision and then we will conduct the next round of review.

We look forward to receiving your revised manuscript.

Kind regards,

Anastassia Zabrodskaja, Ph.D.

Academic Editor

PLOS ONE

Journal Requirements:

3. Please ensure that you include a title page within your main document. You should list all authors and all affiliations as per our author instructions and clearly indicate the corresponding author.

Reviewers' comments:

Reviewer's Responses to Questions

**Comments to the Author**

1. Is the manuscript technically sound, and do the data support the conclusions?

Reviewer #1: Yes

2. Has the statistical analysis been performed appropriately and rigorously? 

Reviewer #1: Yes

3. Have the authors made all data underlying the findings in their manuscript fully available?

Reviewer #1: Yes

4. Is the manuscript presented in an intelligible fashion and written in standard English?

Reviewer #1: Yes

5. Review Comments to the Author

Reviewer #1: It is a current trend and good idea to explore the data. Data presentation could be made better by adding most relevant narrations of interviewee as examples. Discussion could be more supported by the previous research studies.

6. PLOS authors have the option to publish the peer review history of their article (what does this mean?). If published, this will include your full peer review and any attached files.

Reviewer #1: No

---

## [Author Response · Author response to Decision Letter 0]

8 Aug 2023

We have uploaded the responde to reviewers as an appendix.

---

## [Decision Letter · Decision Letter 1]

24 Aug 2023

PONE-D-23-08682R1Applying ChatGPT to tackle the side effects of Personal Learning Environments from learner and learning perspective: an interview of experts in higher educationPLOS ONE

Dear Dr. Zheng,

Thank you for submitting your manuscript to PLOS ONE. After careful consideration, we feel that it has merit but does not fully meet PLOS ONE’s publication criteria as it currently stands. Therefore, we invite you to submit a revised version of the manuscript that addresses the points raised during the review process.

ACADEMIC EDITOR: Please make changes according to the reviewers' comments.

We look forward to receiving your revised manuscript.

Kind regards,

Anastassia Zabrodskaja, Ph.D.

Academic Editor

PLOS ONE

Journal Requirements:

Reviewers' comments:

Reviewer's Responses to Questions

**Comments to the Author**

1. If the authors have adequately addressed your comments raised in a previous round of review and you feel that this manuscript is now acceptable for publication, you may indicate that here to bypass the “Comments to the Author” section, enter your conflict of interest statement in the “Confidential to Editor” section, and submit your "Accept" recommendation.

Reviewer #1: All comments have been addressed

Reviewer #2: All comments have been addressed

Reviewer #3: (No Response)

2. Is the manuscript technically sound, and do the data support the conclusions?

Reviewer #1: Yes

Reviewer #2: Partly

Reviewer #3: Partly

3. Has the statistical analysis been performed appropriately and rigorously? 

Reviewer #1: Yes

Reviewer #2: N/A

Reviewer #3: N/A

4. Have the authors made all data underlying the findings in their manuscript fully available?

Reviewer #1: Yes

Reviewer #2: Yes

Reviewer #3: Yes

5. Is the manuscript presented in an intelligible fashion and written in standard English?

Reviewer #1: Yes

Reviewer #2: Yes

Reviewer #3: Yes

6. Review Comments to the Author

Reviewer #1: Author did an effort to incorporate all the suggested changes in manuscript. This could be published by following format procedure of the Journal.

Reviewer #2: There are a few comments from this reviewer.

1) In the revised manuscript, lines 181 & 182, "To collect data, the researchers conducted online semi-structured interviews in 20 Febraury, 2023, using the Tencent Conference tool." It should say "...on 20 February, 2023,...". To correctly use the word 'on' and the spelling for 'February'.

2) Based on the first comment above, how was it possible to interview all nine participants in this study in one day? Are all nine available on that day of interview upon advanced arrangement or appointments for the interview date and different times? Was the interview done simultaneously by all the four authors/researchers and thus was done in a day? Please make the data collection clearer.

3) For 'Section 2.2 Research Instruments' - the terms 'open-ended questionnaire' (line 204), 'The questionnaire' (lines 205 & 210) and 'The survey questions' (line 219) are quite confusing for this qualitative study. The use of questionnaire would imply the respondents wrote or typed their responses in written form. However, it was stated that "The interviews were conducted in English and lasted between 25 to 30 minutes" (lines 183 & 184) and "The interviews were audio-recorded and transcribed for further analysis." (lines 186 & 187). And these subsequent statements informed this reviewer that actual face-to-face interactions or online interactions were conducted with all the nine respondents.

To avoid any further confusions to the readers, this reviewer suggests to remove all the terms referring to questionnaire and survey, and just say "The research instrument used for this study was an interview using ten open-ended interview questions (see Appendix 1). The questions aimed to gather data on the..."

Similarly, to rephrase the statements below accordingly:

"The interview questions was divided into three categories. The first set (Questions 1-3) aimed to understand the experts' definitions of PLE,..."

"The interview questions were reviewed by two experts in the field of Personal Learning Environments..."

Reviewer #3: I appreciate the work put into this new version of the article. However, I think comment 5 is not fully attended. For instance, questions 1-3 do not have any relevant narration included in the section where the responses to these questions are. The discussion section is still not fully supported. I mean there are assertions in the discussion that do not explicitly relate to the findings in the previous section. Is it possible to enhance this section ? e.g. see lines 588-603 how the findings in the literature that you refer relate to your findings...There is not a clear link.

In the discussion there is nothing about the possible role of tutors or academic support in helping learners to build their PLE. No limitations of the study is found either.

In line 425 I think there is a typo after "which means ...." please check and reword if this is the case.

7. PLOS authors have the option to publish the peer review history of their article (what does this mean?). If published, this will include your full peer review and any attached files.

Reviewer #1: No

Reviewer #2: No

Reviewer #3: No

---

## [Author Response · Author response to Decision Letter 1]

19 Sep 2023

We have uploaded the responde to reviewers as an appendix.

---

## [Decision Letter · Decision Letter 2]

3 Oct 2023

PONE-D-23-08682R2Applying ChatGPT to tackle the side effects of Personal Learning Environments from learner and learning perspective: an interview of experts in higher educationPLOS ONE

Dear Dr. Zheng,

Thank you for submitting your manuscript to PLOS ONE. After careful consideration, we feel that it has merit but does not fully meet PLOS ONE’s publication criteria as it currently stands. Therefore, we invite you to submit a revised version of the manuscript that addresses the points raised during the review process.

ACADEMIC EDITOR: Please implement these minor changes.

We look forward to receiving your revised manuscript.

Kind regards,

Anastassia Zabrodskaja, Ph.D.

Academic Editor

PLOS ONE

Journal Requirements:

Reviewers' comments:

Reviewer's Responses to Questions

**Comments to the Author**

1. If the authors have adequately addressed your comments raised in a previous round of review and you feel that this manuscript is now acceptable for publication, you may indicate that here to bypass the “Comments to the Author” section, enter your conflict of interest statement in the “Confidential to Editor” section, and submit your "Accept" recommendation.

Reviewer #1: All comments have been addressed

Reviewer #2: All comments have been addressed

2. Is the manuscript technically sound, and do the data support the conclusions?

Reviewer #1: Yes

Reviewer #2: Yes

3. Has the statistical analysis been performed appropriately and rigorously? 

Reviewer #1: (No Response)

Reviewer #2: Yes

4. Have the authors made all data underlying the findings in their manuscript fully available?

Reviewer #1: Yes

Reviewer #2: Yes

5. Is the manuscript presented in an intelligible fashion and written in standard English?

Reviewer #1: Yes

Reviewer #2: Yes

6. Review Comments to the Author

Reviewer #1: Author did an effort to incorporate all the suggested changes. This could be published by following format procedure of the Journal.

It is good to present all the possible outcomes of the study. However, heading 5 "conclusion and future research" only presents conclusion of the study. It is suggested to write clear future implications of the study or provide clear recommendations for future research.

Reviewer #2: All my comments have been addressed as well as from other reviewers. Good improvement in this revised paper.

7. PLOS authors have the option to publish the peer review history of their article (what does this mean?). If published, this will include your full peer review and any attached files.

Reviewer #1: No

Reviewer #2: No

---

## [Author Response · Author response to Decision Letter 2]

7 Oct 2023

The ethical statement has been integrated at the outset of the Methods section in our manuscript, in line with stipulated requirements.

---

## [Decision Letter · Decision Letter 3]

28 Nov 2023

Applying ChatGPT to tackle the side effects of Personal Learning Environments from learner and learning perspective: an interview of experts in higher education

PONE-D-23-08682R3

Dear Dr. Zheng,

We’re pleased to inform you that your manuscript has been judged scientifically suitable for publication and will be formally accepted for publication once it meets all outstanding technical requirements.

Kind regards,

Anastassia Zabrodskaja, Ph.D.

Academic Editor

PLOS ONE
---

## [Editor Report · Acceptance letter]

7 Dec 2023

PONE-D-23-08682R3 

Applying ChatGPT to tackle the side effects of personal learning environments from learner and learning perspective: An interview of experts in higher education 

Dear Dr. Zheng:

I'm pleased to inform you that your manuscript has been deemed suitable for publication in PLOS ONE. Congratulations! Your manuscript is now with our production department. 

Kind regards, 

on behalf of

Professor Anastassia Zabrodskaja 

Academic Editor

PLOS ONE